# Nanopore Is Preferable over Illumina for 16S Amplicon Sequencing of the Gut Microbiota When Species-Level Taxonomic Classification, Accurate Estimation of Richness, or Focus on Rare Taxa Is Required

**DOI:** 10.3390/microorganisms11030804

**Published:** 2023-03-21

**Authors:** Márton Szoboszlay, Laetitia Schramm, David Pinzauti, Jeanesse Scerri, Anna Sandionigi, Manuele Biazzo

**Affiliations:** 1The BioArte Ltd., SGN 3000 San Gwann, Malta; 2Department of Informatics, Systems and Communication, University of Milan Bicocca, 20126 Milan, Italy

**Keywords:** Nanopore, Illumina, 16S rRNA, gut microbiota, species-level taxonomy

## Abstract

Nanopore sequencing is a promising technology used for 16S rRNA gene amplicon sequencing as it can provide full-length 16S reads and has a low up-front cost that allows research groups to set up their own sequencing workflows. To assess whether Nanopore with the improved error rate of the Kit 12 chemistry should be adopted as the preferred sequencing technology instead of Illumina for 16S amplicon sequencing of the gut microbiota, we used a mock community and human faecal samples to compare diversity, richness, and species-level community structure, as well as the replicability of the results. Nanopore had less noise, better accuracy with the mock community, a higher proportion of reads from the faecal samples classified to species, and better replicability. The difference between the Nanopore and Illumina results of the faecal bacterial community structure was significant but small compared to the variation between samples. The results show that Nanopore is a better choice for 16S rRNA gene amplicon sequencing when the focus is on species-level taxonomic resolution, the investigation of rare taxa, or an accurate estimation of richness. Illumina 16S sequencing should be reserved for communities with many unknown species, and for studies that require the resolution of amplicon sequence variants.

## 1. Introduction

The 16S ribosomal RNA (rRNA) sequence is the best single molecular marker for identifying bacteria, and with the development of high-throughput sequencing methods, 16S rRNA gene amplicon sequencing has become a pivotal tool for the characterization of bacterial diversity and community structure of almost any environment [1,2]. Since the introduction of the MiSeq in 2011, Illumina instruments became the most widely used platforms for 16S rRNA gene sequencing [3]. They offer very low error rates (Phred quality (Q) scores of 30, equivalent to 99.9% accuracy, for the majority of base calls), but are limited in read length, currently capable of reaching only 2 × 300 bp. Furthermore, the results do not have the full information content of the 16S rRNA gene, as the reads can cover only one, two, or (at most) three out of the nine hypervariable regions. This limits the taxonomic accuracy of the results [4]. In addition, the cost of Illumina devices can be challenging for most research groups. Therefore, instead of performing the sequencing runs in-house, it has become common practice to ship DNA extracts or PCR products to sequencing centers. Having to outsource the sequencing limits the possibility of optimizing the library preparation and run parameters for the needs of each research project, possibly resulting in long turnaround times.

A promising solution to this is Nanopore sequencing technology, which has two crucial properties: long reads and a low up-front cost. Firstly, it can sequence the full 16S rRNA gene retrieving the information content of all nine hypervariable regions. Secondly, the cost of a MinION sequencer is a small fraction of that of any Illumina or PacBio sequencing system. This enables individual research groups to set up and optimize their own library preparation and sequencing workflows. In addition, it is a small portable device, allowing the processing of samples on site without the need of elaborate laboratory infrastructure. Consequently, there is growing interest in using Nanopore for 16S rRNA gene amplicon sequencing, and recent publications have demonstrated its success with a variety of samples, for example freshwater [5], seawater and algal surfaces [6], soil [7,8], surfaces in the international space station [9], animal feces [10], urine [11], buccal and rectal swabs [12], ophthalmic samples [13], bronchoalveolar lavage fluid, blood, pleural and ascitic fluid, cerebrospinal fluid, and wound secretions [14].

The most important limitation of Nanopore sequencing is its comparatively high error rate. The benefit provided by the higher information content of the long reads for taxonomic classification is lost if the reads have low accuracy. In 2019, sequencing of mock communities of known composition led to the conclusion that species-level classification was not yet reliable with Nanopore [15]. Later studies determined the sequencing accuracy to be 92.08% [5] and 93.17–95.97% [16]. Considering that the threshold to distinguish bacterial species based on their 16S rRNA sequence has traditionally been 97% similarity and later revised to be 98.65% similarity [17], it is expected that species-level taxonomic classification of Nanopore reads is problematic with such sequencing accuracy rates. However, the introduction of the kit 12 chemistry in late 2021 brought a substantial improvement, with >99% (Q20) accuracy becoming attainable [18]. This makes the question of whether, or when, Nanopore should be adopted as the preferred sequencing technology instead of Illumina for 16S rRNA gene amplicon sequencing ever more relevant.

To obtain a clear answer, a comparison of Nanopore and Illumina sequencing results is needed that: (1) includes samples of known composition, such as a mock community; (2) includes real samples of high bacterial diversity; (3) compares the results on diversity, richness, and community structure instead of focusing only on specific taxa, such as potential pathogens; (4) assesses replicability, i.e., the differences between repeated analyses of the same sample; and (5) preferably uses the same bioinformatical tools and taxonomic reference database to process the Nanopore and Illumina data. Publications that included comparisons between Nanopore and Illumina 16S rRNA gene amplicon sequencing results are listed in Table 1. They fulfilled some of the above points; however, covering all of them in a single study using the current Nanopore sequencing chemistry with improved accuracy is missing. Therefore, using faecal samples collected from six individuals and a mock community sample, we compared 16S rRNA gene amplicon sequencing results from Nanopore and Illumina to assess their ability to distinguish samples based on the bacterial community structure at species and higher taxonomic levels, deliver replicable results and good coverage of species richness. Each sample was sequenced in three technical replicates with both sequencing platforms. The reads were processed with the EzBioCloud MTP pipeline and reference database [19].

## 2. Materials and Methods

### 2.1. Samples

This study included the ZymoBIOMICS Gut Microbiome Standard (Zymo Research, Irvine, CA, USA), hereafter referred to as the mock community, which is a mixture of 14 bacterial, 1 archaeal, and 2 fungal species, as well as faecal samples collected from six individuals: sample A1 from a 2-year-old male child, samples AN1 and S1X from adult females, samples B1 and MD01 from adult males, and sample S2P from an adult male diagnosed with irritable bowel syndrome (IBS). The samples were obtained with the informed consent of the participants or their parents, and were anonymized. The work was carried out from microbial material and no human data was used. The faecal material was sampled with Danastool Sample Collection Microbiome Kit (Danagen, Barcelona, Spain). The samples were homogenized by vigorous shaking and transported to the laboratory at room temperature.

DNA was extracted on the day of sample collection. The Danastool tubes were vortexed and from each sample, a 200 µl aliquot was extracted with MagMAX Microbiome Ultra Nucleic Acid Isolation Kit (Applied Biosystems, Waltham, MA, USA), using a KingFisher Flex Purification System (Thermo Fisher Scientific, Waltham, MA, USA). In the case of the mock community, a 75 µL aliquot was extracted with the same protocol. The DNA concentration in the extracts was determined with the Qubit 1X dsDNA High-Sensitivity Kit (Invitrogen, Waltham, MA, USA).

### 2.2. Nanopore Sequencing

Each DNA extract was processed in three technical replicates. The 16S rRNA gene was amplified with primers 27f (5′-TTTCTGTTGGTGCTGATATTGC-AGRGTTYGATYMTGGCTCAG-3′) and 1492r (5′-ACTTGCCTGTCGCTCTATCTTC-CGGTTACCTTGTTACGACTT-3′). The 22 nt 5′ tails were added to the primers to serve as the annealing sites for the barcoding primers. The PCRs were carried out in 12.5 µL total volume containing 6.25 µL LongAmp Taq 2X Master Mix (New England Biolabs, Ipswich, MA, USA), the primers in 400 nM concentration, and 0.05 ng template DNA. The reactions were run in a T100 thermal cycler (BioRad, Hercules, CA, USA). The initial denaturation at 95 °C for 4 min was followed by 30 cycles of 95 °C for 20 s, 51 °C for 30 s, and 65 °C for 4 min. The final extension was 5 min at 65 °C. The PCR products were checked on a 2% agarose gel and cleaned with 1 × Agencourt AMPure XP (Beckman Coulter, Indianapolis, IN, USA). The purity and yield of the PCR products were assessed with a NanoDrop 8000 spectrophotometer (Thermo Fisher Scientific, Waltham, MA, USA).

Barcodes and adapters were added in a second PCR using primers from the Nanopore PCR Barcoding Expansion Kit EXP-PBC001 (Oxford Nanopore Technologies, Oxford, UK). The reactions contained in 25 µL total volume 12.5 µL LongAmp Taq 2X Master Mix, 0.5 µL barcoding primer mix, and 25 ng 16S rRNA gene PCR product. The reactions were run in a T100 thermal cycler with initial denaturation at 95 °C for 3 min; 12 cycles of 95 °C for 15 s, 62 °C for 15 s, and 65 °C for 4 min; and a final extension for 5 min at 65 °C. The PCR products were checked on a 2% agarose gel, cleaned with 1 × Agencourt AMPure XP, and quantified with a NanoDrop 8000 spectrophotometer.

Since the EXP-PBC001 Kit contains 12 different barcodes, the PCR products had to be split between two libraries, one containing the technical replicates of samples A1, AN1, B1, and the mock community, and the other containing the technical replicates of samples MD01, S1X, and S2P. The libraries were pooled and subjected to DNA repair and end-prep with the NEBNext Companion Module for ONT Ligation Sequencing (New England Biolabs, Ipswich, MA, USA) in a 60 µL volume containing 1 µg DNA. The reactions were incubated in a T100 thermal cycler at 20 °C for 5 min, followed by 65 °C for 5 min. The libraries were cleaned with 1× Agencourt AMPure XP and prepared for sequencing with the Nanopore Ligation Sequencing Kit SQK-LSK112. The two libraries were sequenced on separate R9.4.1 flow cells loading 5 fmol DNA. The flow cells were run on a GridION sequencer with real-time super accurate base-calling in MinKNOW 21.11.7 with Guppy 5.1.13 and a filter threshold of average Q-score 11. Barcoding was completed post-run in MinKNOW with the default settings. The reads are accessible in the European Nucleotide Archive under the accession number PRJEB56380.

### 2.3. Illumina Sequencing

Three aliquots of each DNA extract were sent to Novogene (Cambridge, UK). The V4 region of the 16S rRNA gene was amplified with primers 515f (5′-GTGCCAGCMGCCGCGGTAA-3′) and 806r (5′-GGACTACHVGGGTWTCTAAT-3′), and the PCR products were sequenced on a NovaSeq 6000 generating 250 nt paired-end reads. The reads were demultiplexed and the primer sequences were trimmed by Novogene. The reads were deposited in the European Nucleotide Archive under the accession number PRJEB56380.

### 2.4. Data Processing

Both the Illumina and Nanopore reads were processed with the EzBioCloud Microbial Taxonomic Profiling (MTP) pipeline and the PKSSU4.0 database [19,34] to remove non-target, low-quality, and chimeric reads, and to perform the taxonomic classification of the filtered sequences. Briefly, the pipeline merges paired reads, discards reads that are <100 bp, or >2000 bp, or have an average Q-score < 25, or are not predicted to be 16S sequences. The dataset is then de-replicated and taxonomic assignment is done with VSEARCH [35] against the EzBioCloud 16S database. Reads that do not match any reference sequence are subjected to chimera detection with UCHIME [36] using a manually curated chimera-free reference database. Sequences that do not have a match in the EzBioCloud 16S database with at least 97% similarity are clustered into 97% OTUs with UCLUST [37]. Singletons OTUs are discarded.

From each sample the EzBioCloud server allows uploading up to 100,000 reads to the pipeline which were randomly selected in case more reads were available from a sample. Since the pre-processing steps of the EzBioCloud MTP pipeline were not designed for Nanopore data, it was necessary to trim and filter the Nanopore reads beforehand. Trimming was done with Cutadapt 3.5 [38] to remove adapter, barcode, and primer sequences. The presence of at least 15 bases of each primer sequence was required with no more than 0.2 error. Reads that did not have recognizable primer sequences on both ends were discarded. The filterAndTrim function of the dada2 1.18.0 R package [39] was used to discard reads that were shorter than 1300 bp or longer than 1600 bp, had ambiguous bases, or had more than 25 expected errors.

The results of the EzBioCloud MTP pipeline were organized into three MTP sets: one containing the Nanopore data, one containing the Illumina data, and one merging the data from both sequencing platforms. Hereafter, these are referred to as the Nanopore MTP set, the Illumina MTP set, and the merged MTP set, respectively. The EzBioCloud 16S reference database has different versions to match various regions of 16S sequences. In these, taxa that are indistinguishable by the given 16S region are merged into groups. The merged MTP set was created with a version of the EzBioCloud 16S reference database that joins species that either the V4 or V1–V9 regions can’t distinguish into groups. Consequently, the classification results from the Nanopore data were not identical in the Nanopore MTP set and the merged MTP set. In the analysis of the results, the merged MTP set was used only when it was inevitable to have the Nanopore and Illumina data in a single data matrix.

For comparison, the Illumina reads were also processed with two other pipelines. In the first, the Divisive Amplicon Denoising Algorithm 2 (DADA2) [39] was used to quality filter, trim, denoise, and merge read pairs. Chimeric sequences were then removed using the consensus method. The taxonomic assignment of the amplicon sequence variants (ASVs) was carried out using the feature-classifier2 plugin [40], implemented in QIIME2 against the SILVA SSU non-redundant database (138 release) [41] adopting a consensus confidence threshold of 0.8. In the second pipeline, the reads were processed in mothur [42] following the MiSeq SOP [43] with the RDP taxonomy 18 reference database [44].

### 2.5. Statistical Analysis

The merged MTP set was used only in the cluster analysis and the permutational multivariate analysis of variance (PERMANOVA). All other results were obtained from the Nanopore and Illumina MTP sets. The analyses were carried out in R 4.2.1 (https://www.r-project.org) with functions from the vegan package 2.6-2 [45]. Cluster analysis was performed with the hclust function. The data matrix was relativized before calculating Bray–Curtis dissimilarities. Clusters were found with the unweighted pair group method with arithmetic mean (UPGMA). PERMANOVA based on Bray–Curtis dissimilarities was used with the adonis function to test the marginal effects of sequencing method and the differences between faecal samples. To quantify the proportion of variance explained by these factors, ω^2^ values were calculated from the PERMANOVA results with the adonis_OmegaSq function of the MicEco package 0.9.17 [46]. To compare the differences in bacterial community structure between faecal samples to the differences between technical replicates of the same sample, Bray–Curtis dissimilarities were calculated from the Nanopore and Illumina MTP sets and compared with Welch two-sample t-tests. Shannon and Simpson diversity indices were calculated with the diversity function of the vegan package. Rarefaction curves were generated with the rarecurve function.

## 3. Results

### 3.1. Sequencing Yield

Nanopore sequencing generated 4,289,400 reads in total (Appendix A) with median quality score 17.6 and 17.5 in the two libraries and a median read length of 1637 nt in both libraries. 90.7% of the reads passed trimming and 55.7% (2,387,309 in total) the quality filter corresponding to 113,681 ± 16,289 reads per sample. Illumina sequencing yielded 2,509,425 read pairs in total or 119,496 ± 11,038 reads per sample. The read length was 253 nt and 93% of the bases had quality scores >30. From each sample, up to 100,000 Nanopore reads and Illumina read pairs were processed with the EzBioCloud MTP pipeline. 1.7% of the Nanopore reads were found to be non-16S, in total 33 reads to be non-bacterial, and 19.8% to be chimeric (Appendix A). 77,438 ± 5496 reads per sample passed the pipeline, 98.7% of which could be classified into species with valid names, phylotypes (putative species represented by full length, high-quality 16S rRNA gene sequences in the taxonomic reference database), or species groups (species that cannot be distinguished based on the sequenced region of the 16S rRNA gene). In comparison, 0.7% of the Illumina reads were low-quality or non-16S, 0.4% non-bacterial, and 8.2% chimeric (Appendix A). 90,817 ± 1481 reads per sample passed the pipeline, 94.5% of which could be classified into species, phylotypes, or species groups.

### 3.2. Mock Community

The mock community contained 18 bacterial strains representing 14 species. With Nanopore sequencing, 35, 38, and 39 species-level taxa (species with valid names, phylotypes, and species groups) were identified in the three technical replicates of the mock community. In contrast, 2076, 2268, and 2548 species-level taxa were found in the Illumina data. Due to this gross overestimation of richness, we tried several approaches to remove noise from the Illumina data. Keeping only taxa that were detected in all three technical replicates left 1073 species-level taxa in the dataset, while discarding 4.2%, 5.0%, and 5.2% of the reads in the replicates. Removing species-level taxa that did not have at least 60 reads in total across the three replicates left 195, 195, and 204 species-level taxa but discarded 9.6%, 10.6%, and 12.4% of the reads. Deleting taxa that did not reach at least 0.1% relative abundance in any of the replicates left 87, 88, and 92 species-level taxa but removed 14.7%, 15.3%, and 17.8% of the reads. The Illumina data were also denoised with dada2 in the QIIME2 pipeline which found 949, 1212, and 1411 amplicon sequence variants in the three replicates of the mock community representing 471, 526, and 586 genera. With the mothur MiSeq SOP, 690, 705, and 749 genera were found in the three replicates.

*Clostridium perfringens* and *Enterococcus faecalis* were not detected either with Nanopore or with Illumina sequencing, and the Illumina data also missed *Salmonella enterica* (Table 2). This is explained by the very low abundance of these species in the mock community. In the Nanopore data, *Faecalibacterium prausnitzii* was represented by only a few reads, but many reads were classified into two *Faecalibacterium* phylotypes: *Faecalibacterium* GG697149 and *Faecalibacterium* NMTZ. *Faecalibacterium* GG697149 was also abundant in the Illumina data, as well as *Roseburia cecicola* and *Veillonella dispar*. These are very likely results of the misclassification of *Faecalibacterium prausnitzii*, *Roseburia hominis*, and *Veillonella rogosae* reads into species with very similar 16S rRNA sequences. When these misclassified reads were included, 96.2%, 96.3%, and 96.4% of the Nanopore reads from the three replicates were from species that were part of the mock community (Table 2). Furthermore, nearly 100% of the reads were classified into genera represented in the mock community (Appendix A). In contrast, only 64.7%, 68.4%, and 72.5% of the Illumina reads were classified into the species of the mock community (Table 2). This improved somewhat when the results were classified at higher taxonomic levels: 69.8%, 72.7%, and 76.3% of the Illumina reads of the three replicates were from genera (Appendix A) and 77.8%, 81.0%, and 82.6% from families that were present in the mock community. Removing species that did not reach at least 0.1% relative abundance in any of the replicates increased the proportion of Illumina reads classified into the species of the mock community to 75.9%, 83.2%, and 85.6% (Table 2).

Apart from the misclassification of the *Faecalibacterium prausnitzii* reads, the Nanopore results were close to the composition of the mock community (Table 2). The largest deviations were found in the cases of *Akkermansia muciniphila* and *Clostridioides difficile*, the relative abundances of which were overestimated. The differences between the three replicates were minimal. In contrast, the Illumina results showed large variations between the replicates, especially in the cases of *Bacteroides fragilis*, *Fusobacterium nucleatum*, and *Prevotella corporis*. These differences were also present in the genus-level classification results (Appendix A). The QIIME2 and mothur pipelines gave similar results (Appendix A).

### 3.3. Faecal Samples

With both sequencing methods, nearly 100% of the reads from the faecal samples could be classified to genera- or genus-level phylotypes. At species-level classification, however, 82.1–97.4% of the Nanopore reads but only 30.8–53.8% of the Illumina reads could be assigned to species with valid names or phylotypes (Appendix A). The remaining reads were classified into species groups or could only be classified at a higher taxonomic level.

The Illumina data indicated very high bacterial richness in the faecal samples. It identified 187–292 species, 258–411 species groups, and 791–1236 phylotypes in the samples (Appendix A). In comparison, 82–127 species, 9–32 species groups, and 46–249 phylotypes were found with Nanopore. Similarly, the samples had 353–483 genera according to the Illumina data, but only 63–94 based on Nanopore (Appendix A). However, technical replicates prepared from the same faecal sample were similar to each other in their number of species, species groups, phylotypes, and genera in both the Nanopore and Illumina data. Rarefaction analysis indicated that with Nanopore, our sequencing effort was enough to capture almost the complete bacterial richness in the samples, while with Illumina more sequences would be needed to obtain the same coverage (Appendix A).

Despite the large difference in the number of species-level taxa detected with Nanopore and Illumina sequencing, the difference in Shannon and Simpson diversity was relatively small. In fact, Simpson diversity was higher according to the Nanopore data than based on the Illumina results in samples AN1 and B1 (Figure 1).

The six stool samples were clearly distinguished based on their bacterial community structures in both the Illumina and Nanopore data (Figure 2). The technical replicates prepared from the same faecal sample with the same sequencing method were very similar. In the Nanopore data, the replicates clustered together at shorter distances, indicating less replicate-to-replicate variation than in the Illumina data. In the case of every faecal sample, the technical replicates divided into two clusters according to the sequencing method. This shows that the sequencing method clearly affected the bacterial community structure results, but this effect was smaller than the differences between the six faecal samples. In fact, the difference between the Nanopore and Illumina results was significant (*p* < 0.001) and accounted for 10.9% of the variation in the bacterial community structure at species-level resolution and 10.0% at genus-level resolution. However, the differences between the six faecal samples explained 81.3% and 82.4% of the variation at species- and genus-level, respectively. The clustering results obtained from species- and genus-level taxonomy were very similar (Appendix A).

Differences between the Nanopore and Illumina results were present even at the highest taxonomic level. Illumina identified 46 phyla in the dataset, while Nanopore detected only 10. Regarding the most abundant phyla, the relative abundance of *Firmicutes*, and consequentially *Firmicutes*/*Bacteroidetes* and *Firmicutes*/*Actinobacteria* ratios, were higher with Nanopore than with Illumina in every faecal sample (Table 3). On the other hand, the relative abundance of *Proteobacteria* was consistently measured to be lower with Nanopore. In the case of the other abundant phyla, however, some samples had very similar results in the Illumina and Nanopore datasets (e.g., *Actinobacteria* in sample A1 or *Bacteroidetes* in sample S2P), while others were very different (e.g., *Actinobacteria* in sample S2P or *Bacteroidetes* in sample MD01). The effect of the sequencing method on the results depended on the faecal sample. With the small number of samples included in this study, this interaction impeded the characterization of the differences in the taxonomic composition results between the Nanopore and Illumina data. Nevertheless, it is clear from the data that even at phylum-level classification, the differences between the Nanopore and Illumina results cannot be removed by applying simple conversion factors.

To assess the replicability of the results from the two sequencing methods, we compared the differences in bacterial community structure among technical replicates prepared from the same faecal sample to the differences between the six faecal samples. Bray–Curtis dissimilarities calculated from the Nanopore and Illumina MTP sets showed that with both sequencing methods, the differences in bacterial community structure between technical replicates were minor compared to the differences between the faecal samples (Figure 3). This was the case with both species- and genus-level taxonomy. The difference between the variation among technical replicates from the same sample and the variation among faecal samples was larger in the Nanopore data. Furthermore, the Bray–Curtis dissimilarities between the replicates were significantly smaller with Nanopore than with Illumina (Welch *t*-test *p* < 0.0001 with both species- and genus-level taxonomy). Thus, the replicability was good with both sequencing methods, but better with Nanopore.

## 4. Discussion

The most important concern about Nanopore sequencing is its higher error rate which could undermine the accuracy of the taxonomic classification of the reads. Therefore, our aim was to compare the Nanopore results with the Illumina setup that delivers the highest possible sequencing quality. Hence, we selected the NovaSeq 6000 system and targeted the V4 region of the 16S rRNA gene with which the 2 × 250 bp read pairs have a near complete overlap. This means that most base pairs were sequenced twice, generating an improved consensus sequence. In consequence, hardly any of the Illumina reads had to be removed due to insufficient quality during the data processing. The kit 12 chemistry brought a substantial improvement to the sequence quality of Nanopore; however, a very large portion (44.3%) of the Nanopore reads were still lost at the quality filtering step. With the recently introduced Kit 14 chemistry and R10.4.1 flow cells, it is likely that future studies will not have to expect losing this much data due to insufficient quality. The retained reads, however, had over 98.08% accuracy as estimated from the Q-scores. Since this accuracy is higher than the commonly used 97% similarity threshold to distinguish species, it was sensible to attempt the species-level classification of the reads.

There are no established best practices for the processing of Nanopore 16S rRNA gene amplicon reads and published studies used a wide range of methods and reference databases [47]. We selected the EzBioCloud MTP pipeline as it allowed us to process the Nanopore and Illumina data with the same toolset, except for the primer trimming and initial quality filtering step, and because the EzBioCloud 16S database is especially suitable for the taxonomic classification of full-length 16S reads at species-level. In brief, it is extensive, manually curated, contains only full-length 16S sequences, includes the complete taxonomy of every sequence, and is frequently updated (the current version was released on 7 July 2021). A comparison to the SILVA and GreenGenes [48] databases using mock community data found that the EzBioCloud 16S reference gave better genus- and species-level classification results with lower false-positive and false-negative rates [49]. Furthermore, the EzBioCloud MTP pipeline takes into account which hypervariable regions of the 16S rRNA gene are covered by the reads by merging the sequences in the reference database that are indistinguishable by the given 16S region and labelling them as a species group. Since the Nanopore reads covered all the hypervariable regions of the 16S rRNA gene while the Illumina reads covered only one, these groups were different for the two datasets. For example, with V1 – V9 reads, *Veillonella rogosae* can be distinguished from other *Veillonella* species but with only the sequence of the V4 region it cannot be distinguished from *V*. *parvula*, *V*. *atypica*, *V*. *rodentium*, *V*. *denticariosi*, and *V*. *tobetsuensis* [50]. Thus, when processing V4 reads, these six species are joined under the name “*V. parvula* group” in the EzBioCloud reference database. The merged MTP set that joined the Nanopore and Illumina results into a single data matrix was created with a version of the EzBioCloud 16S reference database that joins species that either the V4 or V1–V9 regions cannot distinguish into groups. If the Nanopore and Illumina results were directly merged without this, several taxa present in the Nanopore data would have been found missing from the Illumina data, not because Illumina sequencing failed to detect their 16S rRNA gene sequences, but because the short reads could not distinguish them from the sequences of related taxa. This would have artificially increased the effect of the sequencing method on the bacterial community structure.

16S rRNA amplicon sequencing results are known to be burdened with noise which may originate from contamination, PCR and sequencing errors, undetected chimeras, and barcode crosstalk. To mitigate the noise and improve the accuracy of the results, removal of singleton and rare OTUs has been recommended [51,52], and denoising methods such as dada2 [39] and UNOISE2 [53] were developed for Illumina reads. Our Illumina results from the mock community showed an extreme overestimation of richness and a large portion of the reads did not classify into taxa present in the mock community, not even at the family level. This indicates that the data had a considerable amount of noise. We arrived at these results with three different data processing pipelines that employ different quality filtering methods, denoising, reference databases, and classification algorithms. Therefore, it is unlikely that the noise was a product of the bioinformatic toolset. Removing rare taxa improved the results, but even applying an abundance threshold that discarded 14.7–17.8% of the data left a 6.5-fold overestimation of species richness and still a large number of reads classified into taxa not present in the mock community. The source of the noise is unclear. Contamination and sequencing errors were not likely to be the cause since there were no detectable products in the negative controls and all run metrics indicated very high sequencing quality. Barcode crosstalk is a possibility as the library included a large number of samples from other clients of the sequencing center. Pooling samples from different projects is common practice, and with the increasing use of large-scale Illumina sequencers, such as the NovaSeq 6000, samples are sequenced in ever-larger libraries, which might make barcode crosstalk a significant source of error. It has been suggested to use the results from a mock community to determine the abundance threshold to discard rare OTUs or taxa and then apply this threshold to remove noise from all the samples [51]. This approach, however, seems questionable in our case. We identified fewer species-level taxa in the Illumina data from the faecal samples than from the mock community replicates, suggesting that the amount of noise was not even across the samples and a filtering threshold that works well for one sample might not be optimal for another. It is possible that low-diversity samples, such as a mock community, are more prone to generating noisy data.

With Nanopore, our results from the mock community were close to its actual composition and consistent between replicates. Over 99.9% of the reads were classified into the genera present in the mock community, indicating that the data were practically noise-free. Not having to apply an abundance threshold to mitigate noise is a major advantage for studies investigating the rare biosphere [54]. It also improves the performance of richness estimators that rely on the number of singleton and doubleton OTUs or taxa such as the Chao1 [55] and ACE [56] indices or breakaway [57]. The *F. prausnitzii* AP34BHI strain that was present in the mock community has six copies of the 16S rRNA gene in its genome which cluster into two groups by similarity, one consisting of two copies and the other of four copies (Appendix A). This ratio is similar to the F. GG697149:F. NMTZ relative abundance ratios in our results which were 1:2.19, 1:2.02, and 1:2.11 in the three replicates. Thus, it is likely that the intra-genomic variation in the *F. prausnitzii* AP34BHI 16S rRNA gene sequence led to the classification error.

In the case of the faecal samples, both sequencing methods delivered replicable results and could clearly distinguish the samples based on their bacterial community structures. The results with Nanopore and Illumina were not identical, but the effect of the sequencing method on the bacterial community structure was small compared to the differences between faecal samples. Differences between the Nanopore and Illumina results were present at every taxonomic level. However, these can be attributed to the different primer pairs and PCR protocols, the effects of which on 16S rRNA gene sequencing results are well documented in the literature [58,59]. Therefore, our investigation focused on the power to distinguish faecal samples and the replicability of these results instead of identifying differentially abundant taxa.

The Nanopore data outperformed the Illumina data with respect to the proportion of reads from the faecal samples that could be classified to species. This highlights the improved taxonomic resolution of the long reads that has been previously shown with in silico analysis and PacBio sequencing [4]. Due to the better taxonomic resolution, we expected that the Nanopore reads would identify a larger number of species in the faecal samples than the short Illumina reads. Yet, more species were found in the Illumina data. The fact that the samples had somewhat fewer Nanopore than Illumina reads might explain a small part of the difference in species richness between the two datasets. However, the rarefaction curves showed that the coverage of the Nanopore data was much better despite the lower read counts. The true richness of the samples is unknown, but based on the results from the mock community, we suspect that the Illumina data overestimated the number of species in the faecal samples due to noise.

In this study, Nanopore 16S rRNA gene amplicon sequencing showed good accuracy and good replicability at species-level taxonomic resolution. Thus, the data quality provided by the current sequencing chemistry and flow cells is clearly sufficient for characterizing the species composition of complex bacterial communities, such as the human gut microbiota. Methods developed to further decrease the error rate by amplicon self-ligation and rolling circle amplification [60], or with using molecular tags to build consensus sequences [61], are not required to achieve this. However, they might still be necessary if the aim is to reach the resolution of strains and amplicon sequence variants. A novel pipeline, Emu [62], is also a promising tool to correct errors and improve the accuracy of Nanopore 16S rRNA gene amplicon sequencing, but as the expectation–maximization algorithm inflates the number of rare species and requires the application of an abundance threshold, Emu is only applicable when rare taxa and the estimation of richness and diversity are not in the focus.

To our knowledge, this is the first study to utilize the same bioinformatics pipeline and reference database to compare the accuracy of species-level classification of a mock community, the obtained diversity, richness, and community structure of real samples, and the technical replicability of Oxford Nanopore and Illumina technology simultaneously. Our results show that Nanopore long-read sequencing makes up for its higher error rate for species-level resolution. In conclusion, Nanopore is a better choice than Illumina for 16S rRNA gene amplicon sequencing of the gut microbiota when the focus is on assessing bacterial community structure at species-level taxonomic resolution, on the investigation of rare taxa, or on obtaining an accurate estimation of richness. Using Illumina 16S sequencing should be reserved for bacterial communities with many unknown species that are thus not represented in reference databases, and for studies that require the resolution of amplicon sequence variants. With the frequent release of improved sequencing chemistries and flow cells, the advantages of Nanopore can only be expected to grow in the near future.

## Figures and Tables

**Figure 1 microorganisms-11-00804-f001:**
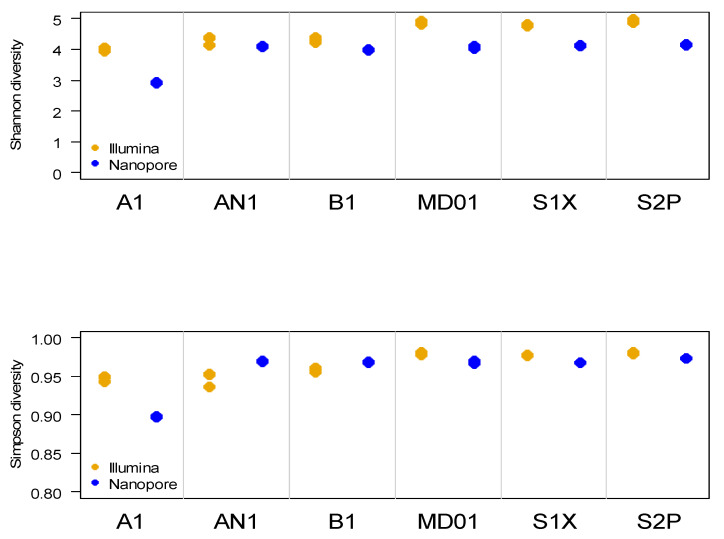
Shannon and Simpson diversity in the faecal samples.

**Figure 2 microorganisms-11-00804-f002:**
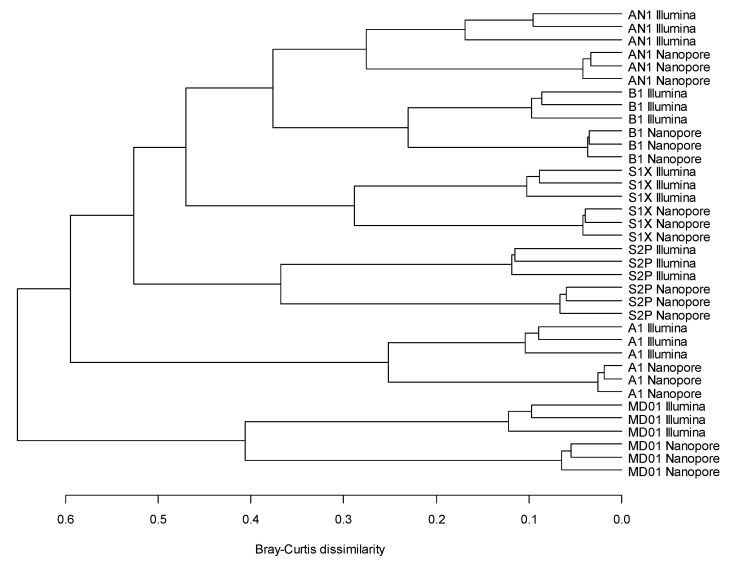
Clustering of the faecal samples based on their species-level bacterial community structure.

**Figure 3 microorganisms-11-00804-f003:**
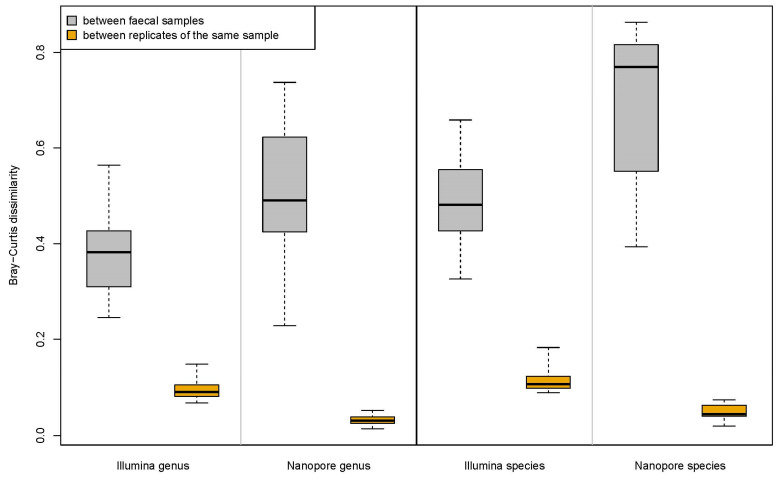
Dissimilarity in bacterial community structure at species- and genus-level taxonomy between technical replicates of the same sample and different faecal samples. The bold lines show the median values, the boxes cover the first to the third quartiles, and the whiskers extend to the data extremes.

**Table 1 microorganisms-11-00804-t001:** Published studies comparing 16S rRNA gene amplicon sequencing with Illumina and Nanopore.

Samples	Mock Community Included	Nanopore Sequencing	Illumina Sequencing	Difference in Community Structure	Difference in Diversity	Difference in Replicability	References
2 faecal samples from mice	No	V1-9, SQK-MAP006	V3-4	No difference, except at species level	Not compared	Not assessed	[20]
13 water samples	Yes	V1-9, SQK-RAB204	V4	Focus on the detection of potential pathogens and faecal indicators	Not compared	Not assessed	[21]
1 mock community	Yes	V1-9, SQK-RAB201	V1-3, V3-4, V4, V4-5, V4-6, V6, V8, V8-9	Good results with both at genus level	Not compared	Not assessed	[15]
6 tap water and biofilm samples	Yes	V1-9, SQK-RAB204 or SQK-RAB201	V3-4	Only the 15 most abundant genera were considered	Not compared	Not assessed	[22]
59 human nasal swabs	No	V1-9, SQK-RAB201	V5-6	Similar results	Similar diversity	Not assessed	[23]
12 indoor dust samples	No	V1-9, SQK-LSK108	V3-4	Difference at genus and species level, but not at family level	Not compared	Not assessed	[24]
50 human faecal samples	No	V1-9, SQK-16S024	V3-4	Good correlation at genus level but not at species level	Not compared	Not assessed	[25]
Only a mock community was included in the comparison with Illumina	Yes	V1-9 and 16S-ITS-23S, SQK-PBK004	V3-4	Nanopore performed better at genus level	Not compared	Only assessed for 16S-ITS-23S	[26]
Corneal and conjunctival swabs from 2 patients	No	V1-9, kit not specified	V4	Similar results	Not compared	Not assessed	[27]
6 human faecal samples	Yes	V1-9 and V3-4, SQK-PBK004	V3-4	Only the 15 most abundant genera were considered	Not compared	Not assessed	[28]
8 human endometrial tissue samples	No	V1-9, SQK-RAB204	6 primer pairs pooled to cover V1-9	Similar results	Not compared	Not assessed	[29]
8 mock communities	Yes	V1-9, SQK-LSK108	V1-2, V1-3, V3, V4	Nanopore showed overall higher bias	Not compared	Not assessed	[30]
31 surface swabs from intensive care units	No	V1-9, SQK-16S024	V4	Focused on the detection of potential pathogens	Not compared	Not assessed	[31]
A mixture of DNA from 72 cheek skin swabs	Yes	V1-9 and 16S-ITS-23S, SQK-LSK110	V1-3	Similar results	Not compared	Good correlation between duplicates	[32]
8 soil samples from 2 sites	Yes	V1-9 and V3-4, SQK-LSK109	V3-4	Significant effect, but not as strong as the differences between sites	Difference in richness estimates	Not assessed	[33]

**Table 2 microorganisms-11-00804-t002:** The relative abundance (%) of bacterial species in the mock community compared to the Nanopore and Illumina results.

Species	Composition	Nanopore			Illumina Unfiltered			Illumina Filtered		
*Akkermansia muciniphila*	0.97	1.61	1.68	1.58	0.67	0.58	0.39	0.79	0.69	0.47
*Bacteroides fragilis*	9.94	13.97	14.21	13.88	7.76	6.85	12.60	9.10	8.09	15.33
*Bifidobacterium adolescentis*	8.78	6.66	7.20	6.92	8.11	9.54	8.17	9.51	11.27	9.93
*Clostridioides difficile*	2.62	5.40	5.05	5.41	1.69	2.70	2.15	1.98	3.19	2.61
*Clostridium perfringens*	0.0002	0	0	0	0	0	0	0	0	0
*Enterococcus faecalis*	0.0009	0	0	0	0	0	0	0	0	0
*Escherichia coli group*	12.12	8.72	8.79	8.51	5.41	8.42	7.53	6.34	9.94	9.16
*Faecalibacterium prausnitzii*	17.63	0.0035	0.0048	0.0039	11.56	15.94	15.69	13.55	18.83	19.08
*Fusobacterium nucleatum*	7.49	7.57	7.33	7.03	9.08	13.60	4.47	10.65	16.06	5.44
*Lactobacillus fermentum*	9.63	5.67	5.65	6.00	2.30	3.11	2.25	2.69	3.68	2.73
*Prevotella corporis*	4.98	6.57	6.42	6.47	5.55	0.94	4.14	6.51	1.11	5.03
*Roseburia hominis*	9.89	7.10	7.07	7.06	4.08	5.49	4.79	4.79	6.48	5.83
*Salmonella enterica*	0.009	0.018	0.024	0.012	0	0	0	0	0	0
*Veillonella rogosae*	15.87	18.44	18.69	19.36	2.69	1.71	1.94	3.16	2.02	2.35
Total	99.9	81.7	82.1	82.2	58.9	68.9	64.1	69.1	81.4	78.0
*Faecalibacterium* GG697149		4.39	4.54	4.42	1.44	0.93	1.15	1.68	1.09	1.40
*Faecalibacterium* NMTZ		9.62	9.15	9.31	0	0	0	0	0	0
*Roseburia cecicola*		0.043	0.043	0.046	1.43	1.1	1.12	1.68	1.29	1.36
*Veillonella dispar*		0.45	0.43	0.39	2.92	1.58	2.02	3.43	1.87	2.46
Total	99.9	96.2	96.3	96.4	64.7	72.5	68.4	75.9	85.6	83.2

The Illumina filtered results were obtained by removing taxa that did not reach at least 0.1% relative abundance in any of the replicates.

**Table 3 microorganisms-11-00804-t003:** Relative abundance (% ± SD) of the dominant phyla and their ratios in the faecal samples.

Sample	Sequencing	*Firmicutes*	*Bacteroidetes*	*Actinobacteria*	*Proteobacteria*	*Verrucomicrobia*	*Firmicutes*/*Bacteroidetes*	*Firmicutes*/*Actinobacteria*
A1	Nanopore	79.1 ± 0.3	7.6 ± 0.1	12.5 ± 0.4	0.5 ± 0.01	0.25 ± 0.04	10.3 ± 0.15	6.4 ± 0.23
	Illumina	71.6 ± 1.3	9.5 ± 0.3	12.2 ± 1.0	4.6 ± 0.2	0.20 ± 0.03	7.5 ± 0.36	5.9 ± 0.57
AN1	Nanopore	64.9 ± 0.2	14.2 ± 0.3	18.8 ± 0.2	2.2 ± 0.2	0.0004 ± 0.0008	4.6 ± 0.06	3.5 ± 0.06
	Illumina	50.2 ± 1.2	19.0 ± 1.6	23.5 ± 1.8	4.3 ± 0.2	0.39 ± 0.05	2.6 ± 0.25	2.2 ± 0.23
B1	Nanopore	73.1 ± 0.4	10.6 ± 0.2	15.5 ± 0.4	0.8 ± 0.1	0.012 ± 0.002	6.9 ± 0.17	4.7 ± 0.10
	Illumina	62.4 ± 0.6	11.4 ± 0.8	18.6 ± 0.8	5.8 ± 0.4	0.22 ± 0.03	5.5 ± 0.46	3.4 ± 0.12
MD01	Nanopore	77.8 ± 0.8	1.7 ± 0.1	11.3 ± 0.6	3.3 ± 0.1	5.05 ± 0.17	46.7 ± 2.05	6.9 ± 0.46
	Illumina	60.9 ± 2.4	12.7 ± 0.3	14.8 ± 1.1	4.2 ± 0.5	4.55 ± 1.68	4.8 ± 0.26	4.1 ± 0.51
S1X	Nanopore	66.9 ± 0.8	23.1 ± 0.6	8.7 ± 0.3	0.7 ± 0.02	0.095 ± 0.02	2.9 ± 0.10	7.7 ± 0.31
	Illumina	58.3 ± 1.1	21.6 ± 0.8	11.0 ± 0.4	6.2 ± 0.2	0.42 ± 0.02	2.7 ± 0.15	5.3 ± 0.26
S2P	Nanopore	66.9 ± 1.0	28.5 ± 0.6	1.0 ± 0.2	3.7 ± 0.3	0.0012 ± 0.0012	2.4 ± 0.06	71.5 ± 12.23
	Illumina	53.8 ± 1.2	26.8 ± 2.0	6.7 ± 0.6	9.9 ± 0.7	0.55 ± 0.07	2.0 ± 0.20	8.1 ± 0.56

## Data Availability

All sequencing data are accessible in the European Nucleotide Archive under the accession number PRJEB56380.

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
