# Peer review of "Nanopore Is Preferable over Illumina for 16S Amplicon Sequencing of the Gut Microbiota When Species-Level Taxonomic Classification, Accurate Estimation of Richness, or Focus on Rare Taxa Is Required"

_microorganisms, 2023, doi:10.3390/microorganisms11030804_

Round 1

Reviewer 1 Report

This research highlights the usefulness and benefits of third-generation sequencing technology (using nanopores) over second-generation Illumina NGS (using reversible dye terminators) in identifying 16S rRNA in diverse bacterial communities.

I suggest this paper to be accepted after minor revision.

The introduction provides a solid, comprehensive overview of the issue that enables the reader to easily grasp the breadth of this technology's uses. The references are current and pertinent to the subject. In particular, the authors explore taxonomic classification challenges at the species level.

Despite the fact that nanopore sequencing has a known accuracy of 92-97% whereas Illumina sequencing has a known accuracy of 99.9%, the objectives in line 73 are well defined and realistically address all aspects in order to determine if nanopore gives a clear advantage over Illumina.

Material and Methods are stated effectively; nevertheless, I believe a few additional species-specific characteristics should be included in section 2.1 (if I'm not mistaken, these are supplied in Table 2?). The protocols for nanopore and Illumina sequencing, as well as the data pipeline setup and statistics utilized, are satisfactory. The experimental setup is relatively typical and appropriate for the study, given that the primary purpose of the paper is not to invent a novel approach, but rather to demonstrate and compare the efficacy of the two examined techniques.

I suggest shortening discussion a bit at lines 434-457.

Conclusion is concise and straightforward: Nanopore is superior to Illumina for 16S rRNA gene amplicon sequencing of the gut microbiota when analyzing bacterial community structure at species-level taxonomic resolution, investigating uncommon taxa, or obtaining an accurate estimate of richness is the objective. This is the fundamental reason why researchers should employ this method.

Author Response

Thank you for your comments and suggestions! They were appreciated and considered, and the manuscript was updated accordingly.

Section 2.1 describes the sampling and DNA extraction. The composition of the mock is indeed in Table 2, which is referenced in section 3.2 where the results from this mock sample are described. Regarding the faecal samples, we do not have metadata to add to section 2.1 that we could correlate with species-specific characteristics in the results. This would be difficult with such low number of samples and would be beyond the scope of this study.

The section in lines 434-457 was revised to make this part of the discussion shorter and more straightforward.

Reviewer 2 Report

Thank you for inviting me to review the manuscript titled “Nanopore is preferable over Illumina for 16S amplicon sequencing of the gut microbiota when species-level taxonomic classification, accurate estimation of richness, or focus on rare taxa is required”.

This manuscript compared the difference between the Nanopore and Illumina results of the faecal bacterial community structure and the variation between samples. The results showed that Nanopore is a better choice when the focus is on species-level taxonomic resolution, and an accurate estimation of richness. In total, I am not excited when I read through all the results, since most of the advantages Nanopore technology contributes are well studied by literatures, e.g.:

·      Heikema, Astrid P., et al. "Comparison of illumina versus nanopore 16S rRNA gene sequencing of the human nasal microbiota." Genes 11.9 (2020): 1105. [The authors included it in reference section, but should discussed more to distinguish their own contributions]

·      Chen, Zhao, David L. Erickson, and Jianghong Meng. "Benchmarking hybrid assembly approaches for genomic analyses of bacterial pathogens using Illumina and Oxford Nanopore sequencing." BMC genomics 21.1 (2020): 1-21.

·      Egeter, Bastian, et al. "Speeding up the detection of invasive bivalve species using environmental DNA: A Nanopore and Illumina sequencing comparison." Molecular Ecology Resources 22.6 (2022): 2232-2247.

I was surprising that the author did not cover any comparison about the read-length comparison between the two platforms in their experiments and results. This will be a good point to show evidence of variant calling and haplotype detections of Nanopore data (reported by lots of literatures).

Other issue draws my attention is which basecaller used for Nanopore sequencing data? This will influence the accuracy reported in results sections.

I would recommend the authors update their experiments with latest basecaller Guppy to check accuracy compared to Illumina platform.

There is no experimental comparison of the two platforms distinguished with previously known work. The topic and the application proposed is not so novel. I will reject the current version.

Author Response

Thank you for your comments and suggestions! They were appreciated and considered, and the manuscript was updated accordingly.

The benefit of the long reads for the taxonomic classification is indeed expected but the error rate of Nanopore sequencing has only now reached the point that accurate species level classification is possible without employing laborious techniques such as rolling-circle amplification and molecular tags. The goal of our study was to demonstrate this and assess the accuracy and replicability of the method. This information is novel and valuable for research groups that need 16S amplicon sequencing data with better taxonomic resolution.

The study of Heikema et al. (10.3390/genes11091105) is briefly described in Table 1. We don’t see merit in a detailed comparison to our results as the two studies had different aims. Heikema at al. investigated the human nasal microbiota in paediatric patients which is a community with extremely low diversity (4.5 genera on average) and focused on genus level classification. This was necessary as the Nanopore sequencing chemistry at the time had high error rates not suitable for calling species. In fact, they concluded that “At the species level, it appears that advances still need to be made to improve the accuracy of taxonomic classification by nanopore sequencing (as with other sequencing technologies). Since our initial comparative studies began, accurate taxonomic assignment at species level using nanopore sequencing continues to improve, with advances in reducing the relatively high error rate of nanopore sequencing, generating obvious advantages. Such changes are to be welcomed.” Our aim was to assess if the development of Nanopore sequencing has reached the point that it allows species level classification of the reads; and, as opposed to Heikema et al., we included testing the accuracy of the results with a mock community of known composition, and the replicability with technical replicates.

Chen et al. (10.1186/s12864-020-07041-8) focused on the assembly of shotgun genomics data from bacterial strains. Since our manuscript is on a very different scenario: 16S amplicon sequencing to describe the structure of diverse microbial communities at species-level resolution; referencing Chen et al. would not bring much to the discussion.

Egeter et al. (10.1111/1755-0998.13610) used amplicon sequencing to detect 8 potentially present bivalve species in lakes. They targeted 137 and 128 bp long loci; thus, they took no advantage of long Nanopore reads. The bioinformatic approach they applied is quite specific to their aims and methods. Our study had very different goals and methodology; therefore, we don’t see the benefit of including the study of Egeter et al. in the discussion.

We showed that the accuracy of Nanopore already allows species level classification of the 16S reads. However, despite the development of the flow cells, sequencing chemistry, and basecallers, we find that the sequencing quality is still not high enough to investigate single nucleotide variations in amplicon sequencing data. We discussed this in lines 473 – 485. Our dataset is not suitable for variant calling and haplotype detection.

We used Guppy 5.1.13. This information was added to section 2.2. This was the latest basecaller version available at the time of the data analysis.

Reviewer 3 Report

The titled should be changed, and the content should be improved largely. "Nanopore is preferable over Illumina for 16S amplicon sequencing of the gut microbiota when species-level taxonomic classification, accurate estimation of richness, or focus on rare taxa is required".If we want to identify "species-level " ,mostly we need full-length 16S, however, the authors compared the Illumina for 16S amplicon sequencing and Nanopore. I think there is not sufficient enough to support the scientific meaning. In my view, both methods have advantages, we need how to use these method not to "preferable " but tell ths facts which environment these method should be and could be used.

Author Response

Thank you for your comments and suggestions! They were all appreciated and considered, and the manuscript was updated accordingly.

In this study, we assessed the accuracy and replicability of describing the bacterial community structure at species level taxonomic resolution with Nanopore 16S rRNA amplicon sequencing and compared it with using Illumina for the same purpose. Indeed, both sequencing methods have advantages, which we compared and discussed. Based on this, we arrived at recommendations on under which circumstances is Nanopore preferable over Illumina for 16S rRNA amplicon sequencing. Due to the long reads, there is a large interest in using Nanopore for 16S sequencing but, as reported in previous literature, up to this point the error rate of Nanopore sequencing made species level classification of the reads problematic without employing laborious techniques such as rolling-circle amplification and molecular tags. Our study demonstrates that this is no longer the case and provides detailed information on the accuracy and replicability we can currently expect with Nanopore. This information is novel and valuable for research groups that need 16S amplicon sequencing data with better taxonomic resolution.

Round 2

Reviewer 2 Report

The authors have addressed all my concerns. I am happy with current revision, except Figure 3 (Text is not clear).

I will accept this version after the author correct Figure 3 issues.